# Exploring the Association between Gambling-Related Offenses, Substance Use, Psychiatric Comorbidities, and Treatment Outcome

**DOI:** 10.3390/jcm11164669

**Published:** 2022-08-10

**Authors:** Cristina Vintró-Alcaraz, Gemma Mestre-Bach, Roser Granero, Elena Caravaca, Mónica Gómez-Peña, Laura Moragas, Isabel Baenas, Amparo del Pino-Gutiérrez, Susana Valero-Solís, Milagros Lara-Huallipe, Bernat Mora-Maltas, Eduardo Valenciano-Mendoza, Elías Guillen-Guzmán, Ester Codina, José M. Menchón, Fernando Fernández-Aranda, Susana Jiménez-Murcia

**Affiliations:** 1Department of Psychiatry, Bellvitge University Hospital-Bellvitge Biomedical Research Institute (IDIBELL), 08907 Barcelona, Spain; 2Centro de Investigación Biomédica en Red Fisiopatología Obesidad y Nutrición (CIBERObn), Instituto de Salud Carlos III, 28029 Madrid, Spain; 3Psychoneurobiology of Eating and Addictive Behaviors Group, Neurosciences Programme, Bellvitge Biomedical Research Institute (IDIBELL), 08908 Barcelona, Spain; 4Facultad de Ciencias de la Salud, Universidad Internacional de La Rioja, 26006 La Rioja, Spain; 5Departament de Psicobiologia i Metodologia de les Ciències de la Salut, Universitat Autònoma de Barcelona, 08193 Barcelona, Spain; 6Department of Public Health, Mental Health and Mother-Infant Nursing, School of Nursing, University of Barcelona, 08907 Barcelona, Spain; 7Departament of Child and Adolescent Psychiatry and Psychology, Institute of Neurosciences, Hospital Clinic Barcelona, 08036 Barcelona, Spain; 8Centro de Investigación Biomédica en Red Salud Mental (CIBERSAM), Instituto de Salud Carlos III, 08907 Barcelona, Spain; 9Department of Clinical Sciences, School of Medicine and Health Sciences, University of Barcelona, 08907 Barcelona, Spain

**Keywords:** gambling disorder, gambling-related offenses, dropout, relapse, psychopathology, personality, substance use, psychiatric comorbidity, impulsivity

## Abstract

Several studies have explored the association between gambling disorder (GD) and gambling-related crimes. However, it is still unclear how the commission of these offenses influences treatment outcomes. In this longitudinal study we sought: (1) to explore sociodemographic and clinical differences (e.g., psychiatric comorbidities) between individuals with GD who had committed gambling-related illegal acts (differentiating into those who had had legal consequences (*n* = 31) and those who had not (*n* = 55)), and patients with GD who had not committed crimes (*n* = 85); and (2) to compare the treatment outcome of these three groups, considering dropouts and relapses. Several sociodemographic and clinical variables were assessed, including the presence of substance use, and comorbid mental disorders. Patients received 16 sessions of cognitive-behavioral therapy. Patients who reported an absence of gambling-related illegal behavior were older, and showed the lowest GD severity, the most functional psychopathological state, the lowest impulsivity levels, and a more adaptive personality profile. Patients who had committed offenses with legal consequences presented the highest risk of dropout and relapses, higher number of psychological symptoms, higher likelihood of any other mental disorders, and greater prevalence of tobacco and illegal drugs use. Our findings uphold that patients who have committed gambling-related offenses show a more complex clinical profile that may interfere with their adherence to treatment.

## 1. Introduction

Gambling disorder (GD) is a psychiatric disorder characterized by recurrent and persistent problematic gambling behavior often associated with certain personality traits, cognitive distortions, and co-occurring psychopathology [1,2]. Moreover, GD, similar to other addictions, is characterized by cognitive deficits and alterations in underlying neurobiological mechanisms mainly related to impulsivity, compulsivity, reward/punishment processing, and decision-making [3,4]. GD is leading to clinically significant distress and usually also leads to relevant financial problems [5], which in some cases has been increased in the context of the COVID-19 pandemic [6].

Financial problems arising from GD can lead to the commission of illegal acts, although there is no consensus about the specific causality of this association [7]. Gambling-related crimes are usually committed for two specific purposes: (1) to obtain money to finance the gambling behavior and/or (2) to recoup financial shortfalls resulting from the gambling behavior [8]. Usually, non-violent, income-producing, and property-related offenses are carried out, such as fraud, robbery, forgery, and theft [9,10].

The commission of gambling-related offenses was contemplated as a diagnostic criterion in previous versions of the Diagnostic and Statistical Manual of Mental Disorders (DSM), although in the latest version, the DSM-5 [1], this criterion was eliminated since many authors considered it to be a criterion associated with the severity of the GD, rather than a diagnostic criterion itself [11,12]. Although it is not currently considered a diagnostic criterion, it remains a relevant clinical criterion [13], and numerous research studies have been conducted to explore reasons for which not all individuals engage in gambling-related offenses. Distinct clinical and sociodemographic differences have been identified between individuals with GD who commit illegal acts and those who do not. Some authors have found that committing gambling-related crimes was associated, at the sociodemographic level, with younger age, lower income, and being unemployed [14,15]. At the clinical level, crimes have been linked with greater psychopathology and impulsivity levels, higher GD severity (associated, in turn, with an increased risk of criminal recidivism), earlier GD onset, greater gambling-related debts, and longer duration of the disorder [14,16,17,18,19,20]. In addition, it has been suggested that gambling-related offenses may be a mediating factor between personality traits (such as novelty seeking, for instance) and GD severity [21].

Therefore, those individuals with GD who commit gambling-related illegal behaviors show a clinical profile characterized by a greater severity, which could interfere with GD treatment outcomes. In addition, it has been suggested that substance use and psychiatric comorbidities (e.g., depression, anxiety, and attention-deficit/hyperactivity disorder) may mediate the association between illegal acts and GD [15,22,23].

Ledgerwood et al. [24] observed that those patients with GD who had committed crimes maintained a higher GD severity throughout the cognitive-behavioral treatment (CBT), compared to those who had not committed crimes. However, the treatment outcome of these specific patients has scarcely been explored. Likewise, the commission of offenses, and the specific role of substance use and psychiatric comorbidities have not been explored in depth and there is a paucity of studies that distinguish between those crimes that have entailed legal consequences and those cases where gamblers escaped detection or charge [8]. To address these relevant empirical limitations, the present longitudinal study had two central objectives: (1) to explore sociodemographic and clinical differences between individuals with GD who had committed gambling-related illegal acts (differentiating into those that had had legal consequences and those that had not, and also exploring substance use and psychiatric comorbidities), and patients with GD who had not committed crimes; and (2) to compare the treatment outcome of these three groups, considering dropouts and relapses. We hypothesized that, of the three groups, patients with GD who had committed gambling-related crimes with legal consequences would present a more impaired clinical profile and, consequently, a worse response to treatment.

## 2. Materials and Methods

### 2.1. Participants and Procedure

The sample consisted of 117 consecutive treatment-seeking patients with GD. They were recruited between April 2017 and May 2018 at the Behavioral Addictions Unit within the Department of Psychiatry, at a Spanish University Hospital. They were referred through general practitioners or via other health professionals, such as mental health institutions.

Two face-to-face clinical interviews were conducted by experienced psychologists and psychiatrists before a diagnosis was given. The inclusion criteria were: (1) adult participants (18 years old or more); (b) both genders; (c) sufficient proficiency in Spanish to understand the assessment; and (d) patients who sought treatment for GD as their primary mental health concern and who met DSM criteria for GD. Exclusion criteria included the presence of (1) intellectual disability; (2) an organic mental disorder; (3) a neurodegenerative condition; or (4) an active psychotic disorder. Additional sociodemographic and clinical information was taken through self-report instruments and a specific face-to-face interview was done individually to explore gambling-related illegal acts before initiating outpatient treatment. Participants were classified into three different groups according to their criminal behavior: patients with no history of gambling-related illegal acts (*n* = 85; Illegal−), patients with a history of gambling-related illegal acts without legal repercussions (*n* = 55; Illegal + Cons−), and patients who had committed gambling-related illegal acts that had legal consequences (*n* = 31; Illegal + Cons+). This classification has already been used in previous studies [25]. Only those patients who reported illegal acts on both DSM-IV-TR criterion 8 [26] and the clinical interview were included in the illegal acts groups.

### 2.2. Treatment

The CBT group treatment program received by the participants of the present study consisted of 16 weekly outpatient sessions at our public University Hospital, lasting 90 min each session. The treatment program has already been described elsewhere [27] and it has reported short and medium-term effectiveness [28,29]. The groups were conducted by an experienced clinical psychologist and a licensed co-therapist. The goal of this intervention was to educate patients on how to implement CBT strategies to minimize gambling behavior and eventually obtain full abstinence. The topics addressed in the different sessions included: psychoeducation about GD (its course, diagnostic criteria, risk factors, etc.), cognitive restructuring focused on cognitive distortions (e.g., the illusion of control and magical thinking), stimulus control (money management, self-exclusion programs, avoidance of potential triggers, etc.), emotion-regulation skills training, response prevention, and other relapse prevention techniques.

Throughout the 16 sessions, attendance, control of spending, as well as the occurrence of relapses and dropouts were recorded weekly by the clinical psychologist. In this study, a relapse was understood as the occurrence of a full gambling episode once CBT had begun. This conceptualization is common in many studies assessing patients with GD [28,30]. Failure to attend 3 consecutive sessions was considered a dropout.

### 2.3. Measures


DSM-5M-5 [1]


Patients were diagnosed with pathological gambling if they met DSM-IV-TR criteria [26]. We used DSM-IV-TR criteria because the 8th criterion explores the presence of gambling-related illegal acts. Noteworthy, with the release of the DSM-5 [1], the term “pathological gambling” was replaced with “GD”. All patient diagnoses were post-hoc reassessed and recodified to avoid the confounding effect of increased GD severity in patients with a criminal history. In this regard, only patients who met DSM-5 criteria for GD were included in the present study. The internal consistency in our study sample was α = 0.818.


South Oaks Gambling Screen (SOGS) [31]


The SOGS is a 20-item diagnostic questionnaire that ascertains GD severity. It discriminates between probable pathological, problem, and non-problem gamblers. Both reliability and validity of the Spanish validation of this tool are high [32], and the test–retest reliability (R = 0.98, *p* < 0.01) and internal consistency (Cronbach’s α = 0.94) are excellent. In our study sample, this questionnaire achieved adequate internal consistency (α = 0.734).


Symptom Checklist-Revised (SCL-90-R) [33]


This questionnaire assesses a broad range of psychological problems and psychopathological symptoms. It contains 90 items measuring nine primary symptom dimensions and it also yields a global score (Global Severity Index (GSI)), which is a widely used index of psychopathological distress. The Spanish validation obtained good psychometrical properties, with a mean internal consistency of 0.75 (Cronbach’s alpha) [34]. The internal consistency estimated in the study sample for the global scale was excellent (α = 0.98: α = 0.891 for somatization, α = 0.896 for obsession-compulsion, α = 0.877 for interpersonal sensitivity, α = 0.917 for depression, α = 0.895 for anxiety, α = 0.873 for hostility, α = 0.832 for phobic anxiety, α = 0.798 for paranoid ideation, and α = 0.855 for psychoticism).


Impulsive Behavior Scale (UPPS-P) [35]


This questionnaire assesses 5 dimensions of impulsive behavior through self-report on 59 items: lack of premeditation, lack of perseverance, sensation-seeking, negative urgency, and positive urgency. The Spanish adaptation showed good reliability (Cronbach’s α between 0.79 and 0.93) and external validity [36]. In our sample, internal consistency was α = 0.923: α = 0.854 for negative urgency, α = 0.917 for positive urgency, α = 0.818 for lack of premeditation, α = 0.754 for lack of perseverance, and α = 0.866 for sensation-seeking.


Temperament and Character Inventory-Revised (TCI-R) [37]


It is a 240-item self-reported questionnaire that measures seven personality dimensions: four temperament (novelty seeking, harm avoidance, reward dependence, and persistence) and three character dimensions (self-directedness, cooperativeness, and self-transcendence). We used the Spanish version which showed adequate internal consistency (Cronbach’s alpha α mean value of 0.87) [38]. In the present study, internal consistency was between adequate (α = 0.701 for reward dependence, α = 0.726 for novelty-seeking, α = 0.745 for harm avoidance, and α = 0.772 for cooperativeness) to good (α = 0.819 for self-transcendence, α = 0.846 for self-directedness, and α = 0.862 for persistence).


Other sociodemographic and clinical variables


Additional sociodemographic and clinical variables related to gambling, as well as substance use, and psychiatric comorbidities were assessed by means of a semi-structured face-to-face clinical interview described elsewhere [27]. Socioeconomic status was obtained using the Hollingshead Factor Index, based on the educational attainment and occupational prestige domains [39]. Gambling-related crimes were explored through a face-to-face interview designed for this study by two forensic experts in the field.

### 2.4. Statistical Analysis

Stata17 for Windows was used for statistical analysis [40]. Analysis of variance (ANOVA) was used for the comparison of quantitative variables between the groups, and chi-square tests (χ^2^) for the comparison of categorical variables. For these comparisons, the effect sizes were estimated with the standardized Cohen’s-d for mean differences and Cramer’s-phi (φ) for proportion differences. In addition, Finner’s correction was used to control the potential increase in the Type-I error due to the use of multiple null-hypothesis tests (Finner-method is an alternative procedure to the classic Bonferroni-method) [41].

Kaplan-Meier product-limit estimator was used to obtain the cumulate survival curve for the rate to dropout and relapse, and Long Rank (Mantel-Cox procedure) compared the resulting functions between the groups [42].

### 2.5. Ethics

The present study was carried out in accordance with the latest version of the Declaration of Helsinki. The Research Ethics Committee of Bellvitge University Hospital approved the study, and signed informed consent was obtained from all participants.

## 3. Results

### 3.1. Description of the Sample

Most participants in the study were men (93.0%), with primary (51.5%) or secondary (45.0%) education levels, single (48.0%) or married (37.4%), employed (60.2%), and pertained to mean-low or low socioeconomic levels (91.8%). No statistical differences between groups were found for the sociodemographic variables (see Table 1).

### 3.2. Comparison of the Clinical Profile between the Groups

Table 2 contains the results of the ANOVA comparing the clinical profiles. Patients who reported an absence of gambling-related illegal behavior achieved the oldest mean age, the latest age of onset of gambling-related problems, the lowest GD severity levels (DSM-5 criteria, the SOGS total, and the cumulated debts related to the gambling activity), the most functional psychopathological state (lowest means in the SCL-90-R scales), the lowest impulsivity levels, and a personality profile with the lowest novelty seeking and the highest self-directedness and cooperativeness levels. For patients who reported illegal acts, the presence of legal consequences was associated to higher mean scores in somatization, anxiety, phobic anxiety, and novelty seeking.

Table 3 includes the comparison between the groups for the presence of psychiatric comorbidities and substance use. Compared with the other conditions, the group characterized by the presence of illegal acts without legal consequences achieved higher likelihood of any comorbid mental disorder. The prevalence of other mental disorders different to depression, anxiety, and bipolar disorders was lower within the patients without illegal behaviors. The absence of illegal acts was also related to lower likelihood of substance use, specifically tobacco and illegal drugs.

### 3.3. Comparison of the Therapy Outcomes between the Groups

Table 4 shows the risk of dropout and relapses and the comparison between the groups. For both outcomes, the highest likelihood was associated to the presence of illegal behavior with legal consequences (64.5% of dropout and 32.3% of relapses). Regarding the cumulative survival functions, the patients who reported both illegal behaviors with legal consequences also achieved the highest rate of dropout and relapse during the treatment (Figure 1).

## 4. Discussion

The present study aimed to explore sociodemographic and clinical differences between individuals with GD who had committed gambling-related illegal acts (differentiating into those who had had legal consequences and those who had not), and patients with GD who had not committed crimes. Moreover, we aimed to compare the treatment outcome of these three groups, considering dropouts and relapses.

Regarding sociodemographic factors, the proportion of patients included in the present study was mostly male. This clinical reality supports previous studies, which have highlighted a male-female ratio of individuals with GD of 2.8:1.0 [43]. GD remains, therefore, a disorder more prevalent in men, although it is progressively increasing in women [44,45].

In addition, no differences were found between groups in terms of years of schooling, given that most patients had primary or secondary levels of education and a low or medium-low socioeconomic level. These findings are consistent with previous studies, which also found no differences between patients who had committed illegal acts and those who had not [16,20]. However, they are inconsistent with other research that has highlighted an inverse relationship between education and the risk of committing crimes [46], as well as between social stratification and delinquency [47].

Patients who had committed illegal acts (with or without legal consequences) were younger than those who had not. These findings support the age-of-crime curve, which proposes a bell-shaped pattern in the association between age and crime [48,49]. In adolescence and young adulthood, there would therefore be a greater probability of committing crimes that would subsequently decrease with age. Age was the only sociodemographic factor in which significant differences were found between groups, as occurred in previous studies [24].

Regarding clinical features, patients who had not committed gambling-related illegal acts showed lower GD severity than those who had (with or without legal consequences). Previous studies also reported higher levels of GD severity in those patients who had committed gambling-related crimes [19,25,50,51]. These findings would lend support to the fact that illegal acts are a clear indicator of GD severity, rather than a diagnostic criterion per se [11,12], and that it is unlikely that an individual would commit illegal acts in the absence of other diagnostic criteria for GD [52]. It should be noted, however, that contrary to our hypotheses, no differences in GD severity were observed between the group that had committed illegal acts with legal consequences and the group that had committed them without legal consequences. We had hypothesized a different clinical profile between both groups estimating that those crimes with legal consequences might be more severe than those without legal repercussions. However, it is possible that not having legal consequences does not imply less severity of the crime, but simply that the crime was not detected.

Those patients who had committed gambling-related illegal acts also reported greater levels of impulsivity compared to those who had not. However, no significant differences in impulsivity were detected between individuals who had committed gambling-related illegal acts with or without legal consequences. In this line, previous studies suggested that among the different dimensions of impulsivity contemplated by the UPPS-P model, positive urgency (understood as acting rashly when facing intense positive emotions) and lack of premeditation (defined as the tendency to act without taking into account the possible consequences of the behavior) were predictors of the presence of illegal acts in individuals with GD, and could therefore be considered a risk factor [16].

Furthermore, individuals who had committed illegal acts (and more specifically the group without legal consequences) showed a higher probability of presenting psychiatric comorbidity. These findings are consistent with previous studies, which suggested that comorbid mental disorders may be relevant mediating factors in the association between gambling behavior and crime [22,23]. Moreover, the absence of gambling-related illegal acts was also associated with a lower likelihood of substance use, specifically tobacco and illegal drugs. Previous studies in this line have suggested that the co-occurrence of GD and substance use may enhance a disinhibition effect in the individual, and this may increase the likelihood of committing illegal acts related to gambling [15]. The patients who had not committed gambling-related crimes showed a more adaptive personality profile, with lower novelty seeking and higher self-directedness and cooperativeness levels, compared to those who had committed crimes. These results coincide with previous studies [16], suggesting that especially self-directedness, characterized by greater self-control and skills for achieving goals [37], could to some extent be preventing the commission of illegal acts. In addition, these patients showed lower levels of psychopathology compared to the groups that had committed crimes, as observed in previous studies [20].

Finally, to the best of our knowledge, to date, no study has studied in depth the association between the commission of gambling-related crimes and response to treatment, specifically, dropout and relapse rates. Both dropout and relapse are considered essential to assess GD treatment outcome, along with other variables such as gambling behavior measures (e.g., monthly net expenditure and gambling frequency) and measures of GD-related problems (e.g., social, legal, and financial difficulties) [53]. In the present study, consistent with our hypothesis, the illegal acts with legal consequences group presented a higher risk of both dropout and relapse compared to the other two groups. Therefore, although no significant differences were observed in terms of sociodemographic and clinical factors regarding the presence/absence of legal consequences, it is a relevant factor to consider when analyzing treatment outcomes.

It should be noted that the groups that had committed illegal acts presented a more impaired clinical profile, with greater severity of the disorder and psychopathology, more maladaptive personality traits, and higher levels of impulsivity. All these factors could be interfering with dropout and relapse rates, as previous studies suggest [54,55]. In the specific case of gambling-related offenses, Ledgerwood et al. [24] observed that GD severity was maintained throughout CBT in the group of patients who had committed illegal acts, compared to those who had not. Therefore, the authors suggested that the profile of gamblers with associated offenses might require treatments of longer duration and intensity in order to achieve an effective reduction of GD symptomatology. Gambling-related illegal acts and their legal consequences would therefore be factors to contemplate when analyzing the treatment adherence of this type of patient, as well as when designing treatment programs focused on this specific clinical population.

### Limitations and Future Studies

The present study presents several limitations. First, although an attempt was made to reduce the probability of bias by assessing the commission of gambling-related crimes using two independent clinical interviews (one with DSM criteria and the other specific to illegal acts), both focus on self-reporting, so that failure to disclose these crimes by patients may occur, as previous studies have highlighted [8]. Similarly, psychiatric comorbidity and substance use were self-reported by patients at the initial clinical interview, prior to the beginning of therapy. Therefore, it should be noted that the diagnoses reported may be biased. Second, although the present study reports the presence/absence of legal consequences (a previously unexplored factor), it does not include relevant data associated with criminal behavior, such as the typology of the crime or recidivism. Third, this study included only treatment-seeking individuals, so this may be a more problem-conscious gambler profile. Future studies could also include non-treatment seeking gamblers to contrast the clinical profiles. Fourth, the different clinical factors included (personality, psychopathology and impulsivity) have been evaluated through self-report questionnaires, with their consequent limitations. Finally, although gender is an important factor to take into account in the recovery processes [56], the present study has not explored gender differences.

## 5. Conclusions

Patients who reported an absence of gambling-related illegal behavior were older, and showed the lowest GD severity, the most functional psychopathological state, the lowest impulsivity levels, and a more adaptive personality profile. Patients who had committed offenses with legal consequences presented the highest risk of dropout and relapses, higher number of psychological symptoms, higher likelihood of any other mental disorders, and greater prevalence of tobacco and illegal drugs use. Our findings uphold that patients who have committed gambling-related offenses show a more complex clinical profile that may interfere with their adherence to treatment. Therefore, specific treatment plans are required for this type of patient.

## Figures and Tables

**Figure 1 jcm-11-04669-f001:**
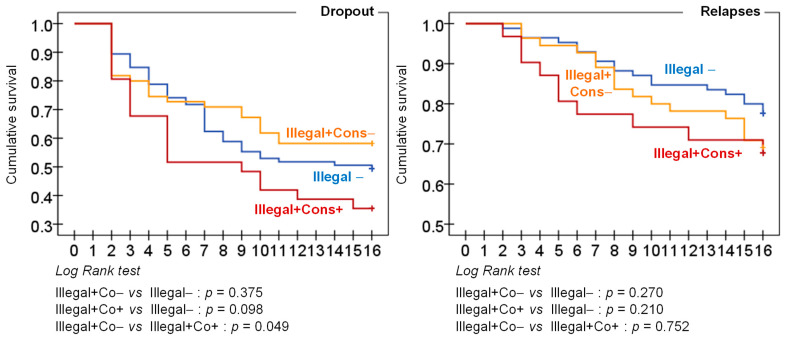
Survival functions for the rate of dropout and relapses. Note. Illegal−: without illegal behavior. Illegal + Cons−: with illegal behavior and without legal consequences. Illegal + Cons+: with illegal behavior and with legal consequences.

**Table 1 jcm-11-04669-t001:** Comparison between the groups for sociodemographic variables.

		Total (*n* = 171)		Illegal− (*n* = 85)		Illegal + Cons− (*n* = 55)		Illegal + Cons+ (*n* = 31)		
		*n*	%	*n*	%	*n*	%	*n*	%	*p*
Gender	Women	12	7.0%	7	8.2%	4	7.3%	1	3.2%	0.643
	Men	159	93.0%	78	91.8%	51	92.7%	30	96.8%	
Education	Primary	88	51.5%	42	49.4%	31	56.4%	15	48.4%	0.740
	Secondary	77	45.0%	41	48.2%	22	40.0%	14	45.2%	
	University	6	3.5%	2	2.4%	2	3.6%	2	6.5%	
Civil status	Single	82	48.0%	33	38.8%	31	56.4%	18	58.1%	0.163
	Married	64	37.4%	39	45.9%	17	30.9%	8	25.8%	
	Divorced	25	14.6%	13	15.3%	7	12.7%	5	16.1%	
Social Index	Mean-high	1	0.6%	1	1.2%	0	0.0%	0	0.0%	0.965
	Mean	13	7.6%	6	7.1%	4	7.3%	3	9.7%	
	Mean-low	66	38.6%	32	37.6%	23	41.8%	11	35.5%	
	Low	91	53.2%	46	54.1%	28	50.9%	17	54.8%	
Employment	Unemployed	68	39.8%	32	37.6%	22	40.0%	14	45.2%	0.764
	Employed	103	60.2%	53	62.4%	33	60.0%	17	54.8%	
Age (years-old); mean-SD		41.38	13.40	45.86	14.00	36.31	10.90	38.10	11.82	**<0.001 ***

Note. Illegal−: without illegal behavior. Illegal + Cons−: with illegal behavior and without legal consequences. Illegal + Cons+: with illegal behavior and with legal consequences. SD: standard deviation. * Bold: significant comparison.

**Table 2 jcm-11-04669-t002:** Comparison between the groups for clinical profiles.

	Illegal− (*n* = 85)		Illegal + Cons− (*n* = 55)		Illegal + Cons+ (*n* = 31)		Illegal + Co− vs. Illegal−		Illegal + Co+ vs. Illegal−		Illegal + Co+ vs. Illegal + Co−	
	Mean	SD	Mean	SD	Mean	SD	*p*	|d|	*p*	|d|	*p*	|d|
Age (years-old)	45.86	14.00	36.31	10.90	38.10	11.82	**<0.001 ***	**0.76 ^†^**	**0.004 ***	**0.60 ^†^**	0.531	0.16
Onset GD (years-old)	31.35	12.62	25.42	8.87	25.18	8.18	**0.003 ***	**0.54 ^†^**	**0.008 ***	**0.58 ^†^**	0.922	0.03
Duration GD (years)	6.05	7.32	5.87	6.09	7.74	6.64	0.883	0.03	0.238	0.24	0.224	0.29
DSM-5 criteria	6.47	2.06	7.53	1.78	7.74	1.73	**0.002 ***	**0.55 ^†^**	**0.002 ***	**0.67 ^†^**	0.619	0.12
SOGS-total	9.69	2.99	11.53	3.21	12.55	3.36	**0.001 ***	**0.59 ^†^**	**<0.001 ***	**0.90 ^†^**	0.149	0.31
Debts (euros)	5757	9943	9914	14,639	9219	14,195	**0.050 ***	0.33	**0.049 ***	0.28	0.744	0.05
SCL-90R Somatization	0.88	0.78	0.93	0.80	1.47	0.90	0.694	0.07	**0.001 ***	**0.70 ^†^**	**0.003 ***	**0.63 ^†^**
SCL-90R Obsessive-comp.	1.05	0.86	1.37	0.88	1.53	0.89	**0.037 ***	0.36	**0.009 ***	**0.55 ^†^**	0.395	0.19
SCL-90R Sensitivity	0.93	0.89	1.13	0.80	1.49	0.83	0.176	0.24	**0.002 ***	**0.65 ^†^**	0.060	0.45
SCL-90R Depression	1.37	0.98	1.69	0.87	2.01	0.92	0.052	0.34	**0.001 ***	**0.68 ^†^**	0.123	0.36
SCL-90R Anxiety	0.93	0.82	1.09	0.70	1.50	0.94	0.246	0.21	**0.001 ***	**0.65 ^†^**	**0.025 ***	**0.50 ^†^**
SCL-90R Hostility	0.77	0.87	1.18	1.00	1.14	0.83	**0.011 ***	0.43	0.056	0.43	0.860	0.04
SCL-90R Phobic anxiety	0.46	0.62	0.51	0.71	0.88	0.89	0.646	0.08	**0.005 ***	**0.55 ^†^**	**0.023 ***	0.45
SCL-90R Paranoia	0.78	0.80	1.12	0.82	1.33	0.88	**0.020 ***	0.41	**0.002 ***	**0.65 ^†^**	0.245	0.25
SCL-90R Psychotic	0.81	0.76	1.05	0.74	1.30	0.91	0.070	0.33	**0.003 ***	**0.59 ^†^**	0.155	0.30
SCL-90R GSI	0.96	0.74	1.19	0.68	1.51	0.78	0.073	0.32	**<0.001 ***	**0.72 ^†^**	0.052	0.44
SCL-90R PST	42.08	23.48	50.51	20.60	57.77	18.84	**0.027 ***	0.38	**0.001 ***	**0.74 ^†^**	0.140	0.37
SCL-90R PSDI	1.82	0.64	2.00	0.54	2.25	0.70	0.101	0.30	**0.001 ***	**0.64 ^†^**	0.077	0.40
UPPS-P Premeditation	23.35	5.23	24.42	6.33	26.26	5.96	0.285	0.18	**0.017 ***	**0.52 ^†^**	0.155	0.30
UPPS-P Perseverance	21.84	4.93	23.13	5.30	23.97	4.19	0.132	0.25	**0.041 ***	0.47	0.449	0.18
UPPS-P Sensation	25.79	7.64	29.93	8.36	29.94	7.73	**0.003 ***	0.52 ^†^	**0.013 ***	**0.54 ^†^**	0.996	0.00
UPPS-P Positive urgency	28.96	7.79	32.60	10.67	34.39	9.69	**0.023 ***	0.39	**0.005 ***	**0.62 ^†^**	0.386	0.18
UPPS-P Negative urgency	30.06	6.55	32.76	8.00	34.97	5.61	**0.025 ***	0.37	**0.001 ***	**0.81 ^†^**	0.157	0.32
UPPS-P Total	129.4	21.40	142.8	24.15	149.6	21.03	**0.001 ***	**0.59 ^†^**	**<0.001 ***	**0.96 ^†^**	0.175	0.30
TCI-R Novelty seeking	106.1	11.96	110.5	13.41	118.0	14.19	0.052	0.34	**<0.001 ***	**0.91 ^†^**	**0.009 ***	**0.55 ^†^**
TCI-R Harm avoidance	101.7	18.91	100.0	17.36	104.3	13.20	0.577	0.09	0.487	0.16	0.281	0.28
TCI-R Reward dependence	99.1	13.47	96.1	14.76	95.0	10.46	0.196	0.21	0.150	0.34	0.727	0.08
TCI-R Persistence	103.7	19.38	106.1	17.27	111.0	16.83	0.463	0.13	0.061	0.40	0.233	0.29
TCI-R Self-directedness	134.2	20.26	123.9	22.10	116.4	20.34	**0.005 ***	**0.52 ^†^**	**<0.0001 ***	**0.88 ^†^**	0.109	0.36
TCI-R Cooperativeness	132.4	16.46	129.7	16.68	124.1	14.83	0.335	0.16	**0.017 ***	**0.53 ^†^**	0.132	0.35
TCI-R Self-transcendence	60.2	13.69	63.4	13.88	66.3	13.27	0.185	0.23	**0.037 ***	0.45	0.348	0.21

Note. Illegal−: without illegal behavior. Illegal + Cons−: with illegal behavior and without legal consequences. Illegal + Cons+: with illegal behavior and with legal consequences. SD: standard deviation. GD: gambling disorder. SOGS: South Oaks Gambling Screen. SCL-90-R: Symptom Checklist-Revised. UPPS-P: Urgency, Premeditation, Perseverance, Sensation Seeking, Positive Urgency. TCI-R: Temperament and Character Inventory-Revised. * Bold: significant comparison. ^†^ Effect size within the range mild-moderate to high-large (|d| > 0.50).

**Table 3 jcm-11-04669-t003:** Comparison between the groups for comorbid mental disorders and substances.

	Illegal− (*n* = 85)		Illegal + Cons− (*n* = 55)		Illegal + Cons+ (*n* = 31)		Illegal + Co− vs. Illegal−		Illegal + Co+ vs. Illegal−		Illegal + Co+ vs. Illegal + Co−	
	*n*	%	*n*	%	*n*	%	*p*	|φ|	*p*	|φ|	*p*	|φ|
Any mental disorder	16	18.8%	18	32.7%	6	19.4%	0.061	**0.158 ^†^**	0.948	0.006	0.184	**0.143 ^†^**
Depression	5	5.9%	4	7.3%	1	3.2%	0.743	0.028	0.568	0.053	0.441	0.083
Anxiety	4	4.7%	4	7.3%	1	3.2%	0.553	−0.050	0.289	0.098	0.450	0.081
Bipolar	3	3.5%	1	1.8%	0	0.0%	0.523	0.054	0.728	0.032	0.441	0.083
Other	3	3.5%	9	16.4%	3	9.7%	**0.008 ***	**0.224 ^†^**	0.186	**0.123 ^†^**	0.390	0.093
Any substance	46	54.1%	38	69.1%	19	61.3%	0.077	**0.149 ^†^**	0.491	0.064	0.463	0.079
Tobacco	41	48.2%	35	63.6%	16	51.6%	0.074	**0.151 ^†^**	0.747	0.030	0.276	**0.118 ^†^**
Alcohol	11	12.9%	6	10.9%	5	16.1%	0.719	0.030	0.659	0.041	0.486	0.075
Illegal drugs	1	1.2%	8	14.5%	5	16.1%	**0.002 ***	**0.266 ^†^**	**0.001 ***	**0.299 ^†^**	0.844	0.021

Note. Illegal−: without illegal behavior. Illegal + Cons−: with illegal behavior and without legal consequences. Illegal + Cons+: with illegal behavior and with legal consequences. |φ|: Phi-statistic. * Bold: significant comparison. ^†^ Effect size within the range mild-moderate to high-large (|φ| > 0.10).

**Table 4 jcm-11-04669-t004:** Comparison between the groups for CBT outcomes.

		Illegal− (*n* = 85)		Illegal + Cons− (*n* = 55)		Illegal + Cons+ (*n* = 31)		Illegal + Co− vs. Illegal−		Illegal + Co+ vs. Illegal−		Illegal + Co+ vs. Illegal + Co−	
		*n*	%	*n*	%	*n*	%	*p*	|φ|	*p*	|φ|	*p*	|φ|
Dropout	Present	43	50.6%	23	41.8%	20	64.5%	0.310	0.086	0.183	**0.124 ^†^**	**0.043 ***	**0.218 ^†^**
	Absent	42	49.4%	32	58.2%	11	35.5%						
Relapses	Present	19	22.4%	17	30.9%	10	32.3%	0.258	0.096	0.276	**0.101 ^†^**	0.897	0.014
	Absent	66	77.6%	38	69.1%	21	67.7%						

Note. Illegal−: without illegal behavior. Illegal + Cons−: with illegal behavior and without legal consequences. Illegal + Cons+: with illegal behavior and with legal consequences. CBT: cognitive-behavioral treatment. |φ|: Phi-statistic. * Bold: significant comparison. ^†^ Effect size within the range mild-moderate to high-large (|φ| > 0.10).

## Data Availability

The datasets analyzed during the current study are not publicly available due to patient confidentiality and other ethical reasons but are available from the corresponding author on reasonable request.

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
