# Peer review of "Exploring the Association between Gambling-Related Offenses, Substance Use, Psychiatric Comorbidities, and Treatment Outcome"

_jcm, 2022, doi:10.3390/jcm11164669_

Round 1

Reviewer 1 Report

The manuscript addresses a very interesting topic and poorly studied in the literature, as the authors mention. It is of special interest to be a study of severity variables that affect the response to treatment in pathological gambling. The theoretical framework is well directed. I especially value the longitudinal design used in the study and the size of the sample, taking into account the difficulty of accessing to this population.

Below I indicate some concerns and minor suggestions in case they are useful for the authors. 

The existence of crimes with and without legal repercussions could be contrasted (in futures studies in this research line) in some more objective way? The testimony of the patient and the assessment of the professionals (two forensic experts) may not be enough, although it is something that the authors honestly already comment in the limitations.

If I have understood the description of the results of table 1 correctly, it is a comparison between the groups for sociodemographic variables and the text states that "no statistical differences between groups were found for the sociodemographic variables" (page 5, line 214-215). However, in section 3.2 where is described the comparison of the clinical profile the text states "patients who reported an absence of gambling-related illegal behavior achieved the oldest mean age..." (Statistical value of the p). I understand that this last data should be in the description of table 1.

In the table 3, interesting information about dropout and relapses is described, being the patients who reported illegal behaviors with legal consequences those who achieved the highest rate of dropout and relapses during the treatment. It is noteworthy that followed by these, those who presented the most dropout were patients had not committed illegal behaviors and finally those with illegal behaviors without legal consequences. Is there any explanation for this that can be added to the discussion?

I am grateful for a study of these characteristics and I congratulate the authors for the excellent paper.  

Author Response

REVIEWER 1

The manuscript addresses a very interesting topic and poorly studied in the literature, as the authors mention. It is of special interest to be a study of severity variables that affect the response to treatment in pathological gambling. The theoretical framework is well directed. I especially value the longitudinal design used in the study and the size of the sample, taking into account the difficulty of accessing to this population.

Reply: We are very grateful for the positive feedback from the reviewer and believe that with their suggestions the article has been greatly improved.

Below I indicate some concerns and minor suggestions in case they are useful for the authors. 

The existence of crimes with and without legal repercussions could be contrasted (in futures studies in this research line) in some more objective way? The testimony of the patient and the assessment of the professionals (two forensic experts) may not be enough, although it is something that the authors honestly already comment in the limitations.

Reply: To our knowledge, only offenses related to gambling behavior that have had legal repercussions could be objectified through court records. However, those offenses without legal consequences could only be recognized by the individuals who have carried them out or by people in their environment who witnessed such offenses. Therefore, as mentioned in the limitations, this is a weak point of this and all studies in this area.

If I have understood the description of the results of table 1 correctly, it is a comparison between the groups for sociodemographic variables and the text states that "no statistical differences between groups were found for the sociodemographic variables" (page 5, line 214-215). However, in section 3.2 where is described the comparison of the clinical profile the text states "patients who reported an absence of gambling-related illegal behavior achieved the oldest mean age..." (Statistical value of the p). I understand that this last data should be in the description of table 1.

Reply: We have included this data in Table 1.

In the table 3, interesting information about dropout and relapses is described, being the patients who reported illegal behaviors with legal consequences those who achieved the highest rate of dropout and relapses during the treatment. It is noteworthy that followed by these, those who presented the most dropout were patients had not committed illegal behaviors and finally those with illegal behaviors without legal consequences. Is there any explanation for this that can be added to the discussion?

Reply: We understand that the reviewer is referring to Table 4. If this is the case, the reviewer's contribution is interesting. However, the differences between the groups commented by the reviewer were not statistically significant. In addition, more balanced sample sizes between the groups would be needed to be able to reach solid conclusions in this regard. This is why we did not want to delve into this secondary aspect in the discussion. However, if the reviewer considers that we need to explain it in depth, we will do so without any problem.

I am grateful for a study of these characteristics and I congratulate the authors for the excellent paper.  

Reply: We are very grateful for the positive feedback from the reviewer.

Reviewer 2 Report

The authors focused on “Exploring the Association between Gambling-related
Offenses, Substance Use, Psychiatric Comorbidities, and Treatment Outcome “.
Research conducted by the authors and the preparation of the manuscript required
scientific knowledge, devoted time and dedication , and all of such factors are
appreciated in high level.
However, it is the duty of the reviewer to define recommendations that the authors
should address in order to increase the value of the manuscript.
Therefore, please take the following reviewer's recommendations into consideration
and apply to all of them:
  • The text of the manuscript, including affiliation names, should be written
    in English only.
  • Line 28-30: Correspondence person's data should include   email address only. Please remove  first and last name, and full postal address.

  • Even though authors exploring the association between gambling-related offenses , it is recommended to update introduction section about gambling behaviours during COVID-19 crisis issues ( in example:Miela, R.J.; CubaÅ‚a, W.J.; Jakuszkowiak-Wojten, K.; Mazurkiewicz, D.W. Gambling behaviours and treatment uptake among vulnerable populations during COVID-19 crisis .Journal of Gambling Issues 2021, Vol. 48, September 23; 233–252 . https://www.scopus.com/record/display.uri?eid=2-s2.0-85120706543&origin=inward&txGid=5497c48b2bf9c0165bf28fbb3d428743&featureToggles=FEATURE_NEW_DOC_DETAILS_EXPORT:1 )

  • Even though authors exploring the association between gambling-related offenses, substance use, psychiatric comorbidities, and treatment outcome; it is recommended to update introduction section about neurobiology of addiction issues ( in example: Miela, R.J.; CubaÅ‚a, W.J.; Mazurkiewicz, D.W.;Jakuszkowiak-Wojten, K. The neurobiology of addiction. A vulnerability/resilience perspective. Review Article. The European Journal of Psychiatry., Volume 32, Issue 3, July–September 2018, pages 139-148 https://doi.org/10.1016/j.ejpsy.2018.01.002

  • A paragraph should be established on the inclusion and exclusion criteria in order to better continue reading scientifically and focus on such issues.

  • Paragraph “5. Conclusions “ must be revised and include more details outcomes of the study performed by authors of this manuscript

  • It is recommended to attach as supplemental materials used in paragraph “2.3. Measures” such as : South Oaks Gambling Screen (SOGS) questionnaire ; Symptom Checklist-Revised (SCL-90-R) questionnaire; Symptom Checklist-Revised (SCL-90-R) questionnaire; Impulsive Behavior Scale (UPPS-P) questionnaire; Temperament and Character Inventory-Revised (TCI-R) questionnaire.

Author Response

REVIEWER 2

The authors focused on “Exploring the Association between Gambling-related Offenses, Substance Use, Psychiatric Comorbidities, and Treatment Outcome”.

Research conducted by the authors and the preparation of the manuscript required
scientific knowledge, devoted time and dedication, and all of such factors are
appreciated in high level.

Reply: We are very grateful for the positive feedback from the reviewer and believe that with their suggestions the article has been greatly improved.

However, it is the duty of the reviewer to define recommendations that the authors should address in order to increase the value of the manuscript.

Therefore, please take the following reviewer's recommendations into consideration and apply to all of them:

The text of the manuscript, including affiliation names, should be written in English only.

Reply: We fully understand the reviewer's suggestion. However, some of the institutions that appear in affiliations and funding explicitly request that their official name not be translated into English, so we cannot make the modification suggested by the reviewer.

Line 28-30: Correspondence person's data should include email address only. Please remove first and last name, and full postal address.

Reply: In accordance with the reviewer's suggestion, we have removed the names and full postal address of the authors of the correspondence.

Even though authors exploring the association between gambling-related offenses , it is recommended to update introduction section about gambling behaviours during COVID-19 crisis issues ( in example:Miela, R.J.; CubaÅ‚a, W.J.; Jakuszkowiak-Wojten, K.; Mazurkiewicz, D.W. Gambling behaviours and treatment uptake among vulnerable populations during COVID-19 crisis .Journal of Gambling Issues 2021, Vol. 48, September 23; 233–252 . https://www.scopus.com/record/display.uri?eid=2-s2.0-85120706543&origin=inward&txGid=5497c48b2bf9c0165bf28fbb3d428743&featureToggles=FEATURE_NEW_DOC_DETAILS_EXPORT:1 )

Reply: We find the reviewer's suggestion interesting, given that COVID-19 has had a strong impact on gambling behaviors. However, our study was not conducted within the framework of COVID-19, so we have not seen fit to add much information about it, so as not to confuse readers. We have added the following information:

“GD is leading to clinically significant distress and usually also leads to relevant financial problems [5], which in some cases has been increased in the context of the COVID-19 pandemic [6].”

Even though authors exploring the association between gambling-related offenses, substance use, psychiatric comorbidities, and treatment outcome; it is recommended to update introduction section about neurobiology of addiction issues ( in example: Miela, R.J.; CubaÅ‚a, W.J.; Mazurkiewicz, D.W.;Jakuszkowiak-Wojten, K. The neurobiology of addiction. A vulnerability/resilience perspective. Review Article. The European Journal of Psychiatry., Volume 32, Issue 3, July–September 2018, pages 139-148 https://doi.org/10.1016/j.ejpsy.2018.01.002

Reply: In accordance with the reviewer's suggestion, we have included the following information:

“Moreover, GD, similar to other addictions, is characterized by cognitive deficits and alterations in underlying neurobiological mechanisms mainly related with impulsivity, compulsivity, reward/punishment processing and decision-making [3,4].”

A paragraph should be established on the inclusion and exclusion criteria in order to better continue reading scientifically and focus on such issues.

Reply: In accordance with the reviewer's suggestion, we have added the following paragraph in the methodology section:

“The inclusion criteria were: (1) adult participants (18 years old or more); (b) both genders; (c) sufficient proficiency in the Spanish language to understand the assessment; and (d) patients who sought treatment for GD as their primary mental health concern and who met DSM criteria for GD. Exclusion criteria included the presence of (1) intellectual disability; (2) an organic mental disorder; (3) a neurodegenerative condition; or (4) an active psychotic disorder.”

Paragraph “5. Conclusions “must be revised and include more details outcomes of the study performed by authors of this manuscript

Reply: We fully agree with the reviewer's suggestion and have included information regarding the main findings of the study. The conclusion section now reads:

Patients who reported an absence of gambling-related illegal behavior were older, and showed the lowest GD severity, the most functional psychopathological state, the lowest impulsivity levels, and a more adaptive personality profile. Patients who had committed offenses with legal consequences presented the highest risk of dropout and relapses, higher number of psychological symptoms, higher likelihood of any other mental disorders, and greater prevalence of tobacco and illegal drugs use. Our findings uphold that patients who have committed gambling-related offenses show a more complex clinical profile that may interfere with their adherence to treatment. Therefore, specific treatment plans are required for this type of patient.”

It is recommended to attach as supplemental materials used in paragraph “2.3. Measures” such as: South Oaks Gambling Screen (SOGS) questionnaire; Symptom Checklist-Revised (SCL-90-R) questionnaire; Symptom Checklist-Revised (SCL-90-R) questionnaire; Impulsive Behavior Scale (UPPS-P) questionnaire; Temperament and Character Inventory-Revised (TCI-R) questionnaire.

Reply: We consider this an interesting suggestion for the reader to become familiar with psychometric instruments. Unfortunately, however, it will not be possible to include the instruments as supplementary material due to copyright issues.

Round 2

Reviewer 2 Report

Reviewer's comments and questions were adequately addressed,
and the manuscript text was revised to a satisfactory level.